# A novel histological index for evaluation of environmental enteric dysfunction identifies geographic-specific features of enteropathy among children with suboptimal growth

Ta-Chiang Liu[1], Kelley VanBuskirk[2], Syed A. Ali[3], M. Paul Kelly[4,5], Lori R. Holtz[2], Omer H. Yilmaz[6], Kamran Sadiq[3], Najeeha Iqbal[3], Beatrice Amadi[4], Sana Syed[3,7], Tahmeed Ahmed[8,9], Sean Moore[7], I. Malick Ndao[2], Michael H. Isaacs[1], John D. Pfeifer[1], Hannah Atlas[10], Phillip I. Tarr[2], Donna M. Denno[10], Christopher A. Moskaluk[11]*

1 Department of Pathology and Immunology, Washington University School of Medicine, St. Louis, MO, United States of America, 2 Department of Pediatrics, Washington University School of Medicine, St. Louis, MO, United States of America, 3 Department of Paediatrics and Child Health, Aga Khan University, Karachi, Pakistan, 4 Tropical Gastroenterology and Nutrition group, University of Zambia School of Medicine, Lusaka, Zambia, 5 Blizard Institute, Barts & The London School of Medicine, Queen Mary University of London, London, United Kingdom, 6 The David H. Koch Institute for Integrative Cancer Research at MIT, Cambridge, MA, Department of Pathology, Massachusetts General Hospital, Boston, MA, United States of America, 7 Department of Pediatrics, University of Virginia, Charlottesville, VA, United States of America, 8 Nutrition and Clinical Services Division (NCSD), International Centre for Diarrhoeal Disease Research, Bangladesh (icddr,b), Dhaka, Bangladesh, 9 James P. Grant School of Public Health, BRAC University, Dhaka, Bangladesh, 10 Departments of Pediatrics and Global Health, University of Washington, Seattle, WA, United States of America, 11 Department of Pathology, University of Virginia, Charlottesville, VA, United States of America

* cam5p@virginia.edu

## Abstract

### Background

A major limitation to understanding the etiopathogenesis of environmental enteric dysfunction (EED) is the lack of a comprehensive, reproducible histologic framework for characterizing the small bowel lesions. We hypothesized that the development of such a system will identify unique histology features for EED, and that some features might correlate with clinical severity.

### Methods

Duodenal endoscopic biopsies from two cohorts where EED is prevalent (Pakistan, Zambia) and North American children with and without gluten sensitive enteropathy (GSE) were processed for routine hematoxylin & eosin (H&E) staining, and scanned to produce whole slide images (WSIs) which we shared among study pathologists via a secure web browser-based platform. A semi-quantitative scoring index composed of 11 parameters encompassing tissue injury and response patterns commonly observed in routine clinical practice was constructed by three gastrointestinal pathologists, with input from EED experts. The pathologists then read the WSIs using the EED histology index, and inter-observer reliability

**Data Availability Statement:** All relevant analyses are within the manuscript and its Supporting

Information files. Primary data (whole slide images) will be made available upon request, as the WSIs are housed in WuPax which is also used as a platform for clinical use. Unrestricted access may cause server to function suboptimally and jeopardize clinical work flow. We do intend to grant access to all request for data, but prefer to be in a controlled fashion so as not to interfere with our clinical work flow. Interested parties can contact data access guarantor Jared Amann-Stewart at amann-stewartj@wustl.edu.

**Funding:** The Pakistan, Zambia, and St. Louis studies were funded by the Bill and Melinda Gates Foundation as independent and distinct research grants (OPP 1144149, 1066200, 1066118, and OPP 1066153, respectively). The funders had no role in study design, data collection and analysis, decision to publish, or preparation of the manuscript.

**Competing interests:** I have read the journal's policy and the authors of this manuscript have the following competing interests: Dr. Phillip Tarr is a consultant for Takeda Pharmaceuticals and the Bill & Melinda Gates Foundation. Dr. Phillip Tarr is also a consultant, member of the Scientific Advisory Board, holder of equity, and a co-inventor on a patent that might earn royalties for MediBeacon Inc. The remainder of the authors have declared that no competing interests exist.

was assessed. The histology index was further used to identify within- and between-child variations as well as features common across and unique to each cohort, and those that correlated with host phenotype.

## Results

Eight of the 11 histologic scoring parameters showed useful degrees of variation. The overall concordance across all parameters was 96% weighted agreement, kappa 0.70, and Gwet's AC 0.93. Zambian and Pakistani tissues shared some histologic features with GSE, but most features were distinct, particularly abundance of intraepithelial lymphocytes in the Pakistani cohort, and marked villous destruction and loss of secretory cell lineages in the Zambian cohort.

## Conclusions

We propose the first EED histology index for interpreting duodenal biopsies. This index should be useful in future clinical and translational studies of this widespread, poorly understood, and highly consequential disorder, which might be caused by multiple contributing processes, in different regions of the world.

### Author summary

The study of EED has been limited by the lack of a rigorously tested, reproducible histology index that can provide insight to the pathogenesis of this entity. In this study we report the first duodenal histology index that was developed using an unbiased approach, with excellent inter-observer reproducibility, for the study of EED. The EED histology index readily identified histologic features that are common or unique to cohorts of distinct geographic locations. Incorporating the histology index into future clinical studies will provide useful insight into the pathogenesis and for intervention strategy development.

## Introduction

Undernutrition remains ubiquitous in low- and middle-income countries (LMICs) and underlies over 40% of child deaths [1]. Wasting, an indicator of acute malnutrition, and defined as weight-for-height standard deviation z score (WHZ) <-2, afflicts 7.3% of children under age five [2]. Severe acute malnutrition (SAM) is defined as WHZ <-3, mid-upper arm circumference <11.5 cm among 6-59-month olds, or the presence of nutritionally-induced bilateral pitting edema. This latter condition is also known as kwashiorkor [3, 4]. Poor linear growth is a reflection of chronic undernutrition. Stunting, defined as height-for-age z score (HAZ) <-2, is used to indicate the prevalence of the problem at the population level. Stunting is associated with increased risk of mortality and developmental delay among children and non-communicable diseases and reduced human capacity into adulthood [1]. An estimated 22% or 150 million children younger than 5 years of age are stunted [2].

Even the most optimal food-based interventions have limited effects on preventing or correcting linear growth deficits [5–8]. Recently, attention has been directed towards environmental enteric dysfunction (EED) as an important component to suboptimal childhood

growth, in particular, poor linear growth [9–13]. EED is an underdiagnosed, highly prevalent condition among children in LMICs [14–18]. EED has been variably termed tropical enteropathy or environmental enteropathy, and was first described as an acquired malabsorption syndrome among adult westerners residing in LMICs [19–21]. A common histologic feature is small intestinal villous blunting, but its etiology remains elusive [22]. The working hypothesis is that an environmental driver, including recurrent or persistent enteric infection, infections from specific pathogen(s), an abundance of nonpathogenic fecal microbes, or distortion of intestinal microbiota composition, could precipitate and/or perpetuate the small bowel response. This process is analogous to the small bowel response to gluten in gluten-sensitive enteropathy (GSE), also known as celiac disease [9, 11, 23].

After a flurry of biopsy studies four decades ago [24–28], there has been a paucity of histologic studies of the small bowel until recently. However, markers in blood, stool, and urine (e.g., urine dual sugar permeability testing) have lent credence to the concept that poor growth and EED are tightly linked in children in LMICs [14, 29–32]. Nonetheless, the origin and pathophysiology of EED remains elusive. This knowledge gap is in part attributable to the limited understanding of the histologic changes associated with this disorder and how they compare to other enteropathies including GSE, which shares features of an environmental trigger and villous blunting on histology [33]. While GSE has not been identified as a prevalent disorder in sub-Saharan Africa and South Asia (outside of India), enteropathy has been identified among children with SAM, especially kwashiorkor, in these settings [34–38].

Based on existing data from cohort studies in diverse populations [14–18], we hypothesized that a histology index using an unbiased approach will allow identification of histopathologic features unique to EED as well as those shared with other enteropathies, and/or identify histologically-identifiable subsets of EED. Here we present the first EED histology index, agnostically constructed and encompassing tissue injury and response patterns commonly observed in clinical practice, in an attempt to improve our understanding of the histopathology underlying this entity or entities. We also describe the rigorous testing used to assess its reproducibility, by applying the scoring index to two distinct cohorts with suboptimal growth recruited from distinct geographic locations where EED is prevalent and North American GSE and control cohorts.

## Methods

### Study cohorts

Endoscopic duodenal biopsies from four cohorts were analyzed: (1) Hospitalized 6–36 month olds with SAM and persistent diarrhea in Lusaka, Zambia [39]; (2) children 18–22 months with non-edematous wasting not requiring hospitalization recruited from a community-based birth cohort in rural Pakistan, and undergoing endoscopy if unresponsive to 2 months of outpatient nutritional intervention [40]; (3) children <18 years with clinically suspected GSE; and (4) children <18 years with normal tissue transglutaminase (tTG) IgA concentrations, no known inflammatory bowel disorders or prior GSE diagnosis, and in whom a diagnostic upper endoscopy was indicated (**Table 1**). The latter two groups underwent endoscopy as part of routine clinical care at the Washington University School of Medicine/St. Louis Children's Hospital (St. Louis, Missouri) Exclusion criteria for Zambian and Pakistani cohorts consisted of neurologic, cardiac, congenital, or other conditions associated with failure to thrive; diagnosis with GSE based on elevated tTG IgA concentrations as also an exclusion criteria for the purposes of this EED histopathology scoring index initiative. Children were excluded from the GSE cohort if they had known inflammatory intestinal disorders or a prior history of GSE.

**Table 1. Cohort inclusion and exclusion criteria.**

|  | Pakistan | Zambia | St. Louis GSE | St. Louis controls |
|---|---|---|---|---|
| **Age criteria for enrollment** | < 18 months | 6–36 months | < 18 years | < 18 years |
| **Age at endoscopy** | 18–22 months | 6–36 months | < 18 years | < 18 years |
| **Inclusion criteria** | Non-edematous wasting not requiring hospitalization recruited from a community-based birth cohort in rural Pakistan, and undergoing endoscopy if unresponsive to 2 months of outpatient nutritional intervention | Hospitalized with SAM and persistent diarrhea | Clinically suspected GSE based on elevated serum concentrations of IgA antibodies to tissue transglutaminase (tTG IgA) and with clinical histopathology results interpreted as GSE | Normal tTG IgA circulating concentrations in whom a decision was made to undergo upper endoscopy |
| **Exclusion criteria** | Neurologic, cardiac, congenital, or other conditions associated with failure to thrive; excluded from present analysis if clinical diagnosis of GSE based on elevated tTG IgA concentrations | Neurologic, cardiac, congenital, or other conditions associated with failure to thrive, or a clinical diagnosis of GSE based on elevated tTG IgA concentrations | Known inflammatory disorders of the gut or a known prior history of GSE | Known inflammatory bowel disorders or prior diagnosis of celiac disease |

## Anthropometric, clinical and laboratory assessments

The following information were collected across all four cohorts: age, sex, recumbent length (children <two years of age), standing height (> two years of age), weight, and history of diarrhea. For convenience, henceforth we refer to both length and height as height. Clinical laboratory testing across all cohorts included hemoglobin, mean corpuscular volume (MCV), C-reactive protein (CRP), and tTG IgA concentrations were determined. tTG IgA assay normal limits differ depending assay kit; we expressed anti-tTG concentrations as percentages of upper limits of normal for the particular kit utilized (IBL International, Germany in Pakistan, Orgentech, Germany in Zambia, Inova, California, and Quest and Labcorp in St. Louis). Additionally, biomarkers potentially reflective of EED or undernutrition processes common to at least two sites' cohorts were analyzed in this report and included urinary dual sugar (lactulose, rhamnose or lactulose, mannitol) permeability testing (Zambia and St. Louis)[41], and serum concentrations of insulin-like growth factor 1 (IGF-1) and glucagon-like peptide 2 (GLP-2) (Zambia and Pakistan). Individual data are included in **S1 Dataset**.

## Duodenal endoscopic biopsy

In the Pakistan cohort, two to three biopsies from the second portion of the duodenum were taken and each biopsy was embedded in an individual paraffin block. In Zambia, typically three biopsies were collected from the second part of the duodenum for each patient. In the St. Louis site, children with suspected GSE had four biopsies taken from the second portion of the duodenum and two from the bulb, while control children had 2 biopsies from the second portion of the duodenum, per that site's clinical protocol [42]. No adverse events were noted during or after the procedure. In Zambia and St. Louis, all biopsies from the same child were embedded in a single paraffin block. In Zambia, all biopsies were collected and oriented under a microscope within 10 minutes before fixation. No additional orientation step was taken before fixation for Pakistani, GSE, and control biopsies.

## Digital pathology workflow

Whole slide images (WSI) were scanned at 20x magnification from hematoxylin & eosin (H&E) stained slides. The slides from the Pakistan cohort were scanned at Aga Khan

University using an Olympus VS120 (Tokyo, Japan) scanner. The slides from the Zambian and St. Louis sites were scanned using an Aperio Scanscope CS scanner at the Washington University Department of Pathology. All WSIs were uploaded to the Washington University Digital Pathology Exchange (WUPax) picture archiving and communication system, a secure browser-based platform developed in 2015 by the Washington University Department of Pathology to meet clinical needs of geographically separated institutions. WUPax is in routine clinical use in a College of American Pathologists (CAP)-accredited, Clinical Laboratory Improvement Act (CLIA)-licensed, and Health Insurance Portability and Accountability Act of 1996 (HIPAA)-compliant environment [43, 44], and has also been adapted for research purposes.

### EED scoring index histology criteria

The EED scoring index was developed by three gastrointestinal pathologists (T.C.L., O.Y., C. A.M.), with input from EED experts who are members of the EED Biopsy Initiative Consortium. This system was developed using a hypothesis-free approach by which common and potential features of small intestinal injury patterns were all included, and scored using semi-quantitative categorical measures)[45]. The three pathologists accessed the WSIs within WUPax. The features included in the scoring index, the definitions of these features, and the scoring options were refined through a series of four iterative reads of subsets of WSIs from the four cohorts. The process also included identifying parameters that merited efforts for improved inter-observer concordance (see Statistical analysis section below). Following these processes, the complete set of WSIs was scored by the three pathologists using the final histology index, and inter-observer concordance was determined. During each set of reads, the pathologists recorded their scores independently and were blinded to cohort identity and each other's scores. A histology atlas was developed (C.A.M.) as a guide for applying the scoring index can be accessed at **S1 Atlas**.

The pathologists scored each WSI as a subjective average across tissue fragments within that WSI. Because the biopsies for the same individual in Zambia and St. Louis were embedded in a single paraffin block, they were therefore processed into a single H&E slide and WSI. However, in Pakistan each biopsy was placed into a single paraffin block, multiple slides and WSIs were available for individual children, enabling us to assess intra-individual score variability between biopsies for this cohort.

### Statistical analysis

Inter-observer agreement was assessed using multiple weighted estimates: percent agreement, kappa, and Gwet's agreement coefficient (AC). Spearman rank correlation was additionally used to assess correlation between pathologists on total histology score, a summed measure of histologic parameters. Statistical differences in histologic parameter scores between cohorts were assessed using the non-parametric Kruskal-Wallis test, appropriate for ordinal dependent variables, followed by Dunn's multiple comparisons test with Benjamini-Hochberg adjustment within tests, with the false discovery rate (FDR) set at 0.05; we did not adjust for FDR across analyses of different histologic score parameters.

Total histologic score percent was utilized in analyses instead of total histologic score to facilitate comparison across images. While the summed total histologic score would be affected by parameters marked as 'not scorable', converting the score into a percent of total possible points (excluding points for non-scorable parameters) eliminates that issue.

Heterogeneity of total histology scores within and between children in the Pakistani cohort, where each WSI represented single biopsies, was analyzed using a cross-sectional mixed effects

model with biopsies nested within each child. The intra-correlation coefficient was used to estimate and compare variances. Averages of scores for individual children were calculated across WSIs for comparative analyses with the other cohorts and relationships with patient characteristics so as not to over-represent one cohort.

Sample sizes were insufficient to conduct hypothesis-driven assessments between histopathology and patient characteristics; however, we did assess for associations as an exploratory analysis. Completeness of anthropometry, but not laboratory, data allowed for cohort-specific assessments of relationship to histology. HAZ, WHZ, body mass index z scores (BMIZ), and weight-for-age z scores (WAZ) were calculated based on World Health Organization (WHO) reference standards [46], using WHO Anthro (version 3.2.2) [47] and WHO Anthro Plus software (version 1.0.4) [48]. Per WHO standards, WAZ was calculated only for children under 10 years of age, and BMI and BMIZ were calculated only for children over age 5 [47]. Relationships between histologic scores and cross-sectional patient characteristics—anthropometrics, clinical parameters (history of diarrhea, HIV infection status), and laboratory markers—were assessed by linear regression with histologic scores treated as continuous predictors after assessment by Akaike information criterion (AIC) and Bayesian information criterion (BIC) in comparison with models using the categorical predictors. Models were adjusted for cohort. We did not adjust for multiple comparisons across regressions because of the exploratory nature of this analysis. Where variation in scores for a histologic parameter was minimal (i.e. representing <2 ordinal levels, with ordinal scoring parameters having either 4 or 5 response levels), we did not assess for a relationship. P-values were two-tailed, and alpha was defined as 0.05.

Statistical analyses were performed using Stata v. 15.0 (StataCorp, College Station, TX) and GraphPad Prism 7.0 (GraphPad Software, San Diego, CA).

### Ethical considerations

Institutional approval was granted by the Aga Khan University Ethical Review Committee (2446-Ped-ERC-13), University of Zambia Biomedical Research Ethics Committee (006-01-13)), and Washington University School of Medicine Human Research Protection Office (201505051). Written consent was obtained from parents for all participants.

## Results

### Demographics and clinical parameters

The demographics of the four cohorts are described in **Table 2**. The 16 children in the Zambian cohort, by study inclusion criteria, were younger (median age 16.5 months) than the Pakistani cohort (median age 22 months). The Zambian children were also, by inclusion criteria, more severely malnourished (e.g., 44% of the Zambian cohort had a WAZ <-4 compared to 10% of the Pakistani cohort) and all had a recent history of diarrhea. Anemia (median hemoglobin 8.4 grams/deciliter [g/dl] and 9.6 g/dl, respectively) and microcytosis (median MCV 70.1 femtoliter [fL] and 62.0 fL, respectively) were common in both cohorts. The Pakistani children and St. Louis controls had CRP circulating concentrations within normal ranges while 31% of Zambian and 22% of the St. Louis GSE cohort had elevated CRP circulating concentrations. Median tTG IgA concentrations as a percent of the upper limit of normal were 5.8% and 5.2% in the Zambian and Pakistani cohorts, respectively.

The St. Louis cohorts (13 GSE and 6 control participants) were older: median age 9.5 and 11.7 years, respectively. One child in each of the St. Louis cohorts had a BMIZ score <-2, and one child in the GSE cohort had both weight-for-age z score (WAZ) and HAZ <-2. All controls had WAZ and HAZ >-2. No GSE participants had a recent history of diarrhea, compared to 33% of controls. The median tTG IgA concentration was 6.8x and 18% of the normal limit

**Table 2. Demographics of the study cohorts.**

| | Pakistan EED | Zambia EED | St. Louis GSE | St. Louis Control |
|---|---|---|---|---|
| | (n = 10) | (n = 16) | (n = 13) | (n = 6) |
| Median age | 22 mos | 16.5 mos | 9.5 yrs | 11.7 yrs |
| IQR | 20–23 | 21-Oct | 5.2–11.7 | 10.2–14.5 |
| (range) | (18–25) | (6–23) | (3.0–15.5) | (2.9–16.9) |
| Sex (% female) | 54.5 | 37.5 | 76.9 | 100 |
| Prevalence of diarrhea N (%) | 5 (50%) | 16 (100%) | 0 | 2 (33.3%) |
| Diarrhea duration (among those with diarrhea) | | | | |
| <14 days | 5 | 0 | - | 0 |
| 14–28 days | 0 | 11 | | 1 |
| >28 days | 0 | 5 | | 1 |
| WHZ score** | | | (<5 yrs age) | (<5 yrs age) |
| > 0 | 0 | 0 | 2 | 1 |
| <0 to >-1 | 1 | 2 | 0 | 0 |
| <-1 to >-2 | 4 | 1 | 0 | 0 |
| <-2 to >-3 | 3 | 5 | 0 | 0 |
| <-3 to >-4 | 2 | 2 | 0 | 0 |
| < -4 | 0 | 6 | 0 | 0 |
| HAZ score | | | | |
| > 0 | 0 | 1 | 9 | 1 |
| <0 to >-1 | 0 | 0 | 2 | 4 |
| <-1 to >-2 | 2 | 2 | 1 | 1 |
| <-2 to >-3 | 3 | 4 | 0 | 0 |
| <-3 to >-4 | 4 | 5 | 1 | 0 |
| < -4 | 1 | 4 | 0 | 0 |
| WAZ score** | | | (<10 yrs age) | (<10 yrs age) |
| > 0 | 0 | 1 | 4 | 1 |
| <0 to >-1 | 1 | 1 | 0 | 0 |
| <-1 to >-2 | 0 | 0 | 2 | 0 |
| <-2 to >-3 | 5 | 0 | 1 | 0 |
| <-3 to >-4 | 3 | 7 | 0 | 0 |
| < -4 | 1 | 7 | 0 | 0 |
| BMIZ score** | n/a | n/a | | |
| > 0 | | | 11 | 4 |
| <0 to >-1 | | | 1 | 0 |
| <-1 to >-2 | | | 0 | 1 |
| <-2 | | | 1 | 1 |
| Median tTG Ab concentration as percentage of upper limit of normal | 5.2 | 5.8 | 681.3* | 18.0* |
| IQR | 4.2–6.4 | 5.2–6.8 | 510.1–1833.3 | 12.5–27.3 |
| (Range) | (2.4–7.2) | (4.8–19.9) | (176.1–4443.3) | (11.0–66.3) |
| Median hemoglobin concentration (g/dl) | 9.6 | 8.4 | 12.7 | 13.5 |
| IQR | 8.1–11.0 | 7.7–8.9 | 11.6–13.2 | 12.9–14.1 |
| (Range) | (7.1–12.4) | (6.1–9.8) | (10.1–14.2) | (12.7–14.1) |
| Missing | 0 | 2 | 3 | 2 |
| Median mean corpuscular volume (fL) | 62 | 70.1 | 80.2 | 83.9 |
| IQR | 56.3–74.4 | 65.6–77.3 | 79.2–83.1 | 78.0–89.7 |
| (Range) | (49.3–82.4) | (61.1–87.0) | (78.8–92.0) | (77.3–90.2) |
| Missing | 0 | 3 | 3 | 2 |

(*Continued*)

**Table 2.** (Continued)

|  | Pakistan EED | Zambia EED | St. Louis GSE | St. Louis Control |
|---|---|---|---|---|
|  | (n = 10) | (n = 16) | (n = 13) | (n = 6) |
| Median C-reactive protein (CRP; ng/ml) | 102 | 1230.8 | 432.6 | 307.9 |
| IQR | 37–199 | 530.4–4261.3 | 163.3–610.2 | 268.7–549.7 |
| (Range) | (14–212) | (47.5–21000) | (40.4–9322.4) | (179.1–1888.1) |
| Missing | 4 | 0 | 4 | 0 |

Abbreviations: Ab, antibody; BMIZ, body mass index z score; CRP: C-reactive protein; EED, environmental enteric dysfunction; fL, femtoliter, GSE, gluten sensitivity enteropathy; HAZ, height-for-age z score; IQR, interquartile range; mos, months; n/a, not applicable; SAM, severe acute malnutrition; tTG Ab, tissue transglutaminase antibody; tTG IgA, tissue transglutaminase immunoglobulin A; WAZ, weight-for-age z score; WHZ, weight-for-height z score; yrs, years.

* Data imputed for two children with GSE and one control based on readings that were imprecise/ out-of-measurement range (>100 and <2, respectively). For children with GSE, we imputed the median of all precise measurements over 100 (n = 5, median 133.3; range 102.0–146.7). For the control child, we imputed the maximum possible value (1.99).

** WHO growth reference standards for WAZ and WHZ scores extend to less than 120 months and BMIZ scores start at 120 months.

for the GSE and control cohorts, respectively. Hemoglobin concentrations were normal among the controls while one child with GSE (representing 10% of GSE children with hemoglobin data) had a hemoglobin <11 g/dl. MCV ranges in both cohorts were normal.

Anthropometric, clinical, and laboratory parameters between the two LMIC study cohorts did not statistically significantly vary except that, per inclusion criteria, the Zambian cohort were younger, had diarrhea of longer duration, and lower WHZ score (S1A–S1C Fig). There was no significant difference in hemoglobin across the cohorts (S1D Fig) and tTG IgA also did not differ between these groups (S1E Fig).

## High inter-observer reproducibility using EED histology scoring index

The histology scoring index assigns a 4–5 tier categorical value to each of the 11 histopathological parameters representing various injury response mechanisms (Table 3). Features with 4-tier grading include: acute (neutrophilic) inflammation, chronic inflammation, eosinophil

**Table 3. Histologic category and scoring scheme.**

| Acute (neutrophilic) inflammation | **0**: PMNs may be present in vessels or in lamina propria, but there is no intraepithelial infiltration (cryptitis, villitis) |
|---|---|
|  | **1**: 1–2 foci of epithelial neutrophilic infiltration or crypt microabscesses |
|  | **2**: > 2 foci of epithelial neutrophilic infiltration or crypt microabscesses, but ≤ 50% of mucosa involved |
|  | **3**: > 50% of mucosa involved by epithelial neutrophilic infiltration |
| Eosinophil infiltration | **0:** No increase in eosinophils (highly scattered in lamina propria, no intravillous or intercryptal space with > 5 eosinophils) |
|  | **1:** Increased eosinophils (intravillous or intercryptal space with > 5 eosinophils) involving ≤ 50% of mucosa, with no eosinophilic crypt microabcesses |
|  | **2:** Increased eosinophils (intravillous or intercryptal space with > 5 eosinophils) involving > 50% of mucosa, or up to 1 focus of eosinophilic epithelial infiltration or crypt microabcesses per mucosal fragment |
|  | **3:** > 2 foci of eosinophilic epithelial infiltration or crypt microabcesses in any mucosal fragment |
| Chronic inflammation | **0:** No qualitative increase in mononuclear inflammatory cells (MIC) in lamina propria. Majority of villous bases contain < 3 MIC across, on average. |
|  | **1:** Increased MIC, based on villous base displaying 3–5 MIC across, on average. |
|  | **2:** Increased MIC, based on villous base displaying 6–10 MIC across, on average. |
|  | **3:** Increased MIC, based on villous base displaying > 10 lymphocytes on average. |

(*Continued*)

**Table 3.** (Continued)

| | |
|---|---|
| **Intraepithelial lymphocytes** | **0**: IEL ratio of lymphocytes to enterocytes < = 20% in any area |
| | **1**: Lymphocyte/epithelial ratio > 20%, but ≤ 50%, in < = 50% of mucosa |
| | **2**: Lymphocyte/epithelial ratio > 20%, but ≤ 50%, in > 50% of mucosa |
| | **3**: Lymphocyte/epithelial ratio > 50% in ≤ 50% of mucosa |
| | **4**: Lymphocyte/epithelial ratio > 50% in > 50% of mucosa |
| **Villous architecture** | **0**: Majority of villi are > 3 crypt lengths long |
| | **1**: Villi are ≤ 3 but > 1 crypt length long, with abnormality in ≤ 50% of mucosa. |
| | **2**: Villi are ≤ 3 but > 1 crypt length long, with abnormality in > 50% of mucosa |
| | **3**: Villi absent, or ≤ 1 crypt length long, with abnormality in ≤ 50% of mucosa |
| | **4**: Villi absent, or ≤ 1 crypt length long, with abnormality in > 50% of mucosa |
| **Intramucosal Brunner glands** | **0**: Brunner glands are in submucosa, but are not observed above the muscularis mucosae |
| | **1**: 1–2 foci, none involving > 5 crypt bases |
| | **2**: 3–5 foci, none involving > 5 crypt bases |
| | **3**: > 5 foci, or any area of intramucosal Brunner glands involving > 5 crypt bases |
| **Foveolar cell metaplasia** | **0**: Only absorptive enterocytes and goblet cells observed on villi, no evidence of foveolar cells |
| | **1**: Foveolar mucin cells observed, usually on the tips of the villi; 1–2 villous tips involved |
| | **2**: Foveolar mucin cells observed, usually on the tips of the villi; 3–5 villous tips involved |
| | **3**: Foveolar mucin cells observed, usually on the tips of the villi; > 5 villous tips involved |
| **Goblet cell density** | **0**: Normal goblet cell density (at least 1 goblet cell per 20 enterocytes) in all evaluable mucosal epithelial layer |
| | **1**: Decreased goblet cells (< 1/20 enterocytes) in 1–25% of evaluable mucosal epithelium |
| | **2**: Decreased goblet cells (< 1/20 enterocytes) in 26–50% of evaluable mucosal epithelium |
| | **3**: Decreased goblet cells (< 1/20 enterocytes) in 51–75% of evaluable mucosal epithelium |
| | **4**: Decreased goblet cells (< 1/20 enterocytes) in 76–100% of evaluable mucosal epithelium |
| **Paneth cell density** | **0**: ≥ 5 Paneth cells/ crypt, on average |
| | **1**: 2–4 Paneth cells/ crypt, on average |
| | **2**: < 2 Paneth cell/crypt, involving ≤ 50% of crypt bases |
| | **3**: < 2 Paneth cell/crypt, involving > 50% of crypt bases |
| **Enterocyte injury** | **0**: Majority of enterocytes (90%) show tall columnar morphology |
| | **1**: Enterocytes show low columnar (< 2:1 L:W ratio), cuboidal or flat morphology, in ≤ 50% of mucosa |
| | **2**: Enterocytes show low columnar (≤2:1 L:W ratio), cuboidal or flat morphology, in > 50% of mucosa |
| | **3**: Any area of mucosal erosion/ulceration |
| **Epithelial detachment** | **0**: Complete coverage of mucosal surface by epithelial cells |
| | **1**: Surface epithelium missing or detached from < 25% of mucosa |
| | **2**: Surface epithelium missing or detached from 25–50% of mucosa |
| | **3**: Surface epithelium missing or detached from 51–75% of mucosa |
| | **4**: Surface epithelium missing or detached from > 75% of mucosa |

Note: For chronic inflammation there is a response option "not scorable due to <3 villi assessable or other slide quality issue", while for all other parameters there is a response option "not scorable due to slide quality or other factors".

infiltration, intramucosal Brunner glands, foveolar cell (gastric mucin cell) metaplasia, Paneth cell density, and enterocyte injury. Features with 5-tier grading include: intraepithelial lymphocytes (IELs), villous architecture, goblet cell density, and epithelial detachment. In addition, a total histologic score is calculated as a sum of the individual parameters, and ranges from 0 to 37. Because individual parameters could potentially be deemed "not determinable" because of slide preparation or other quality issues, we also calculated another summative

score as a percentage (total histologic score percent), to account for instances where the denominator was <37.

There was good agreement across the three pathologists using the final histology index to score the WSIs (Table 4). Chronic inflammation, IELs and villous architecture shared the lowest percent agreement between pathologists (S2A and S2B Fig). Concordance of total histology percent score was 84.0% weighted agreement, kappa 0.41, and Gwet's AC 0.55, and Spearman's correlations for pathologist pairs ranged from 0.58 to 0.74 (S1 Table). The overall concordance across all parameters was 96% weighted agreement, kappa 0.70, and Gwet's AC 0.93. This level of concordance is comparable to those reported in other pathology studies [49–52].

Median total histology scores of the Zambian and Pakistani WSIs fell between those of the control and GSE cohorts (Fig 1 and Table 5; all p-values FDR-adjusted within tests). St. Louis control, Zambian, and Pakistani WSI total histology score percent distributions significantly differed from that of the GSE WSIs (p ≤0.0001, 0.02, and 0.0005, respectively). The distribution of Zambian WSI total score percent differed from those of the non-GSE controls (p = 0.01), while Pakistani WSI score percent did not (p = 0.20). Score percent distributions for Pakistani and Zambian WSIs did not differ significantly (p = 0.13).

Fig 2 (and S2 Table) show the individual histology features for each cohort. Several patterns are discernible. Compared to the GSE WSIs, the Zambian and Pakistani WSIs had less chronic inflammatory infiltration of the lamina propria (Fig 2A), and lower abundances of intramucosal Brunner glands (Fig 2B). However, the Pakistani cohort had markedly increased IELs, with densities similar to those of the GSE cohort (Fig 2C). In contrast, the Zambian cohort had notably fewer secretory lineage cells (goblet and Paneth cells; Fig 2D and 2E), more enterocyte injury (Fig 2F), a tendency toward epithelial detachment (Fig 2G) and more severe villous architectural damage (Fig 2H). There were no substantial neutrophilic inflammation, eosinophilic infiltration, or foveolar metaplasia in tissue from any cohort (S2 Fig). Pathogens were only observed in the Pakistani cohort and limited to *Giardia* in five of 10 children's slide images (two of whom had *Giardia* detected in all of the WSI).

## Heterogeneity of pathology findings in the Pakistani cohort

A distinguishing feature of the Pakistani study design is that each biopsy was allocated to single slides, thereby yielding multiple distinct WSIs per child for analysis of lesion patchiness. As

**Table 4. Concordance measures of EED duodenal histology index.**

| Histology feature | % agreement | Fleiss' Kappa | Gwet's AC |
|---|---|---|---|
| Acute (neutrophilic) inflammation | 99.2 | -0.01 | 0.99 |
| Eosinophil infiltration | 96.2 | 0.12 | 0.96 |
| Chronic inflammation | 68.9 | -0.09 | 0.39 |
| Intraepithelial lymphocytes | 76.4 | 0.32 | 0.45 |
| Villus architecture | 77.0 | 0.41 | 0.45 |
| Intramucosal Brunner glands | 90.8 | 0.74 | 0.83 |
| Foveolar cell metaplasia | 99.2 | -0.01 | 0.99 |
| Goblet cell density | 84.5 | 0.21 | 0.72 |
| Paneth cell density | 81.4 | 0.48 | 0.60 |
| Enterocyte injury | 85.4 | 0.20 | 0.76 |
| Epithelial detachment | 86.2 | -0.02 | 0.78 |
| Total score percent | 84.0 | 0.41 | 0.55 |
| Overall concordance | 96.0 | 0.70 | 0.93 |

Cohort-specific histology features

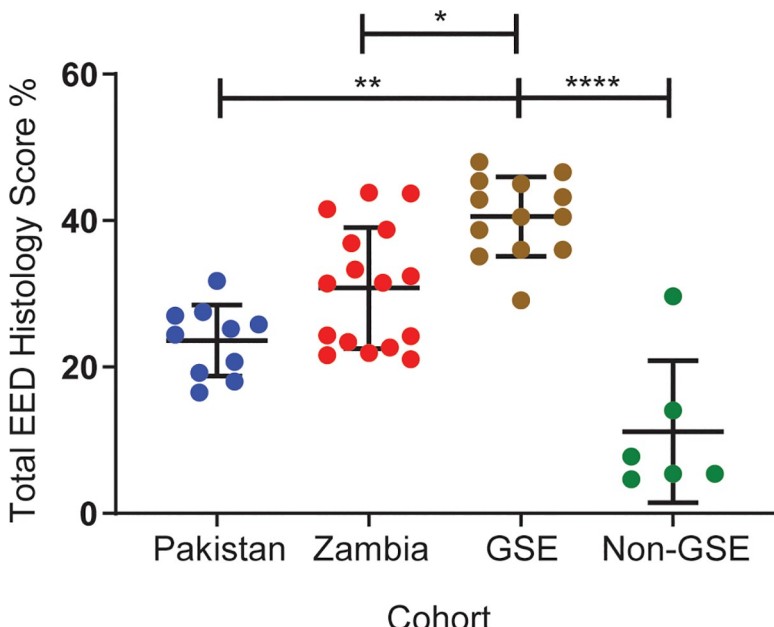

**Fig 1. Total histology score percent, by cohort.** Compared to the St. Louis GSE cohort, the St. Louis control, Pakistani and Zambian cohorts had lower total histology scores (p<0.0001, p = 0.0005, and p = 0.02, respectively). Total histology score percent did not differ significantly between the Pakistani and Zambian WSIs (p = 0.13). Statistical analysis was performed by Kruskal-Wallis test followed by Dunn's multiple comparisons test with Benjamini-Hochberg correction for false discovery error rate. *: p ≤ 0.05; **: p ≤ 0.01; ****: p ≤ 0.0001. The bars indicate mean ± standard deviation.

seen in **Fig 3A**, the maximal discrepancy between different biopsies from the same patient could generate a difference in total histologic percent score of ~5. Individual histologic features that tended to differ by ≥2 levels of pathology between biopsies from the same child include villous architecture (**Fig 3B**) and the presence of intramucosal Brunner glands (**Fig 3C**). In contrast, IEL, goblet cell and Paneth cell densities as well as chronic lamina propria inflammation were often only 1–2 levels apart between different biopsy specimens from an individual (**S3 Fig**). Estimates of intra-class correlation from a multilevel model indicated that within-child variation exceeded between-child variation for total histologic score percent (57.4% vs. 42.6%, respectively, p = 0.013).

**Table 5. Median total histology score and total histology score percent by cohort.**

| Parameter (range of possible scores) | Pakistan | Zambia | St. Louis GSE | St. Louis Controls |
|---|---|---|---|---|
| Median total score (0–37) | 9.3 | 11.5 | 15 | 2.3 |
| IQR | 7.1–10.0 | 8.3–13.8 | 13.3–16.3 | 2.0–5.0 |
| Range | 6.1–11.7 | 7.3–15.3 | 9.7–17.3 | 1.7–10.3 |
| Median total score percent (0–100)[1] | 25.2 | 31.5 | 40.5 | 6.6 |
| IQR | 19.2–27.5 | 23.0–37.9 | 36.0–45.0 | 5.4–14.1 |
| Range | 16.5–31.8 | 21.1–43.8 | 29.1–48.0 | 4.7–29.7 |
| p value[2] compared to GSE | 0.0005 | 0.02 | - | <0.0001 |

Abbreviations: GSE, gluten sensitive enteropathy; IQR, interquartile range.

[1] Compared to the GSE control group, Zambian WSI total score percent differed significantly (p = 0.01) while Pakistani WSI score percent did not (p = 0.20).

Differences between Pakistani and Zambian cohorts were not significant (p = 0.13).

[2] FDR-adjusted (within test) p-values.

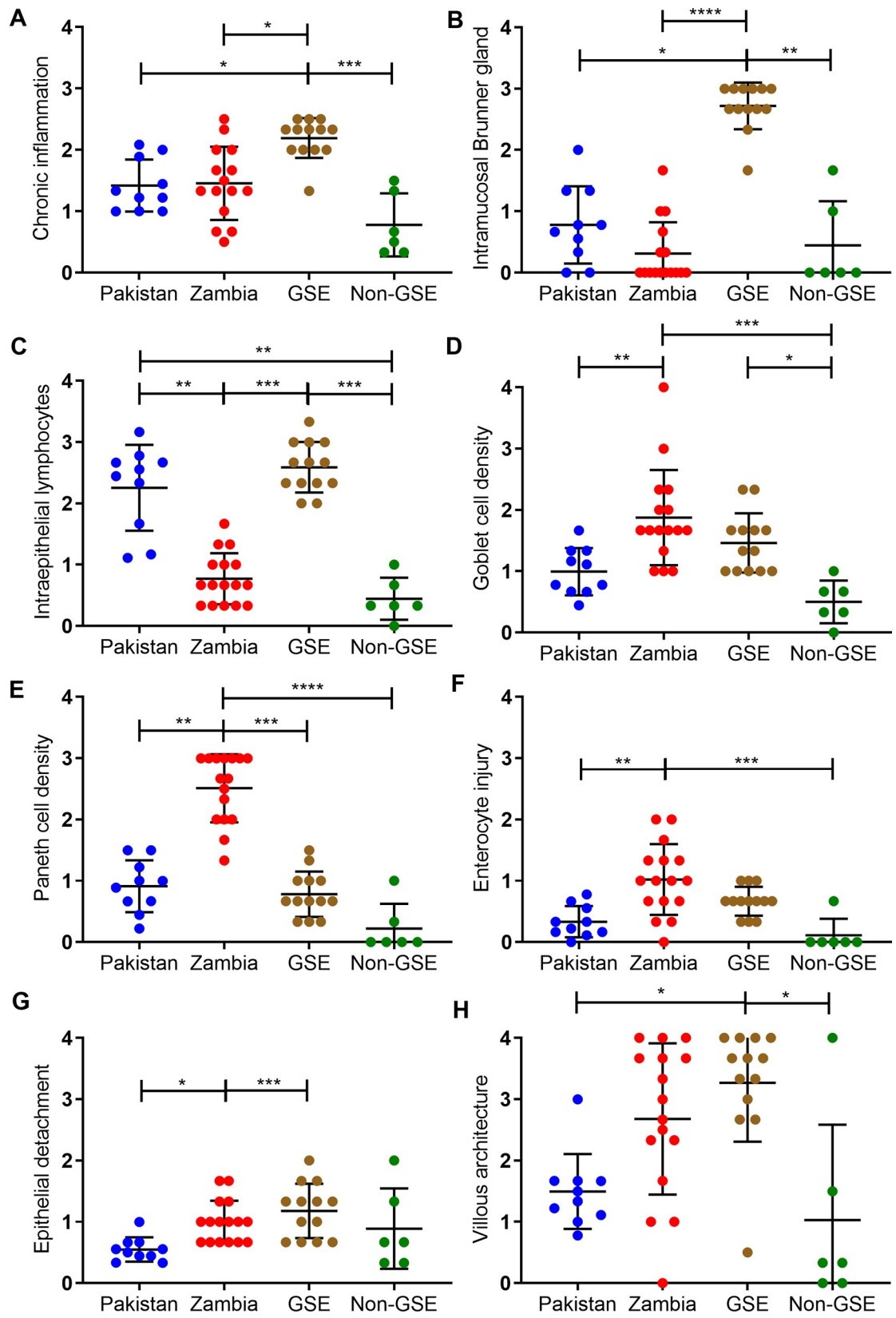

**Fig 2. Scores of individual histology features, by cohort.** The GSE cohort had the highest scores in the categories of (A) chronic inflammation and (B) intramucosal Brunner's gland. Between the Pakistani and Zambian cohorts, the Pakistani cohort had (C) more intraepithelial lymphocytes. In contrast, the Zambian cohort had higher scores in the categories of (D) goblet cell density, (E) Paneth cell density, (F) enterocyte injury, and (G) epithelial detachment. The Zambian cohort also tended to have higher scores in (H) villous architecture derangement. Sample size: Pakistani n = 10, Zambian n = 16, GSE n = 13, control n = 6. *: $p \leq 0.05$; **: $p \leq 0.01$; ***: $p \leq 0.001$; ****: $p \leq 0.0001$. Statistical analysis was performed by Kruskal-Wallis test followed by Dunn's multiple comparisons test and corrected within tests for false discovery rate by the Benjamini-Hochberg method. The bars indicate mean ± standard deviation.

To determine if the reading of individual versus multiple biopsies may have affected the scoring reproducibility by the study pathologists, we took advantage of their assessment of the number of tissue fragments on each WSI. The cohort was divided into two sets, one in which the average number of tissue fragments was one (n = 15) and the other in which the average was >1 (n = 46). The degree of agreement between the pathologists for the histologic parameters was approximately the same (**Table 6** and **S2 Dataset**), indicating that the number of tissue fragments being assessed by the pathologists did not significantly contribute to the degree of histologic agreement.

### Association of histology scores to patient characteristics

We next determined if the histology scores correlated with clinical characteristics. HIV infection status in the Zambia cohort was not associated with total histology percent score or any individual histology parameter scores. After adjusting for site, total histologic score percent and the histology parameter scores were not significantly associated with HAZ, or WHZ in the combined study cohort. The only site-specific associations were identified in the Pakistani cohort, for which IEL density was inversely associated with all three anthropometric parameters: HAZ (ß = -1.1, 95% CI -1.8, -0.5; p = 0.004), WAZ (ß = -1.1, CI -2.1, -0.1; p = 0.03), and WHZ (ß = -0.8, CI -1.4, -0.2; p = 0.01). Also, in this cohort an unexpected positive association was observed between villous architecture score and LAZ (ß = 1.2, CI 0.4, 2.0; p = 0.008).

Adjusting for cohort, clinical laboratory and EED or growth biomarkers were similarly not significantly associated with histology total score percent or subscores, with one exception: IGF-1, measured in the EED cohorts only, was borderline associated with Paneth cell density (ß = -8.0, CI-15.7, -0.4; p = 0.04). No associations were found between histologic scores and lactulose: mannitol ratios (not performed in the Pakistani cohort, and half of the GSE and control subjects were tested for lactulose: rhamnose ratio), hemoglobin concentrations, MCV, CRP, tTG IgA (expressed as a percent of upper limit of normal) or GLP-2 concentrations. Several marginally significant relationships between histology scores and patient characteristics of small effect sizes were observed (**S3 Table**).

### Discussion

Patterns of histologic changes in the gut have long been associated with both specific diarrheal and malabsorptive entities, as well as non-specific reactive/adaptive changes to injury. This includes histologic features that have been identified among individuals from indigenous populations and from travelers spending time in "tropical" areas that have experienced chronic malabsorption syndromes, so-called "tropical sprue" [53]. Similar histologic changes have been noted in asymptomatic individuals in the same geographic areas, so-called" "tropical enteropathy", now referred to as environmental enteropathy or as EED [10, 54]. Among the issues that have hampered identification and subclassification of EED and the discovery of its underlying etiologies has been the lack of a robust and systematic method of categorizing and quantifying the intestinal histologic changes from cohorts of affected individuals.

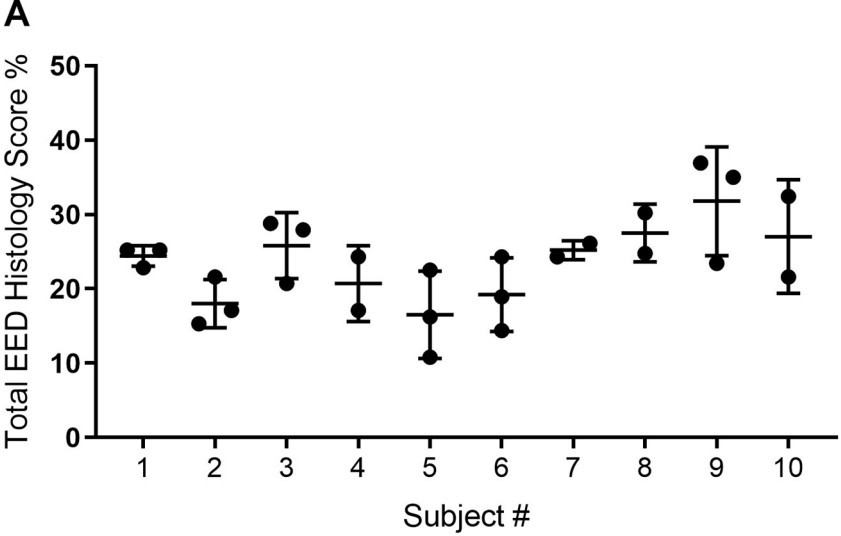

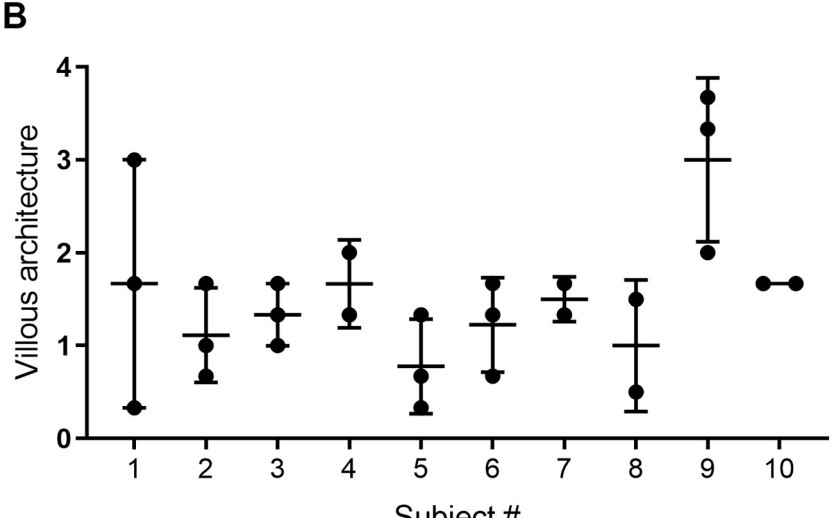

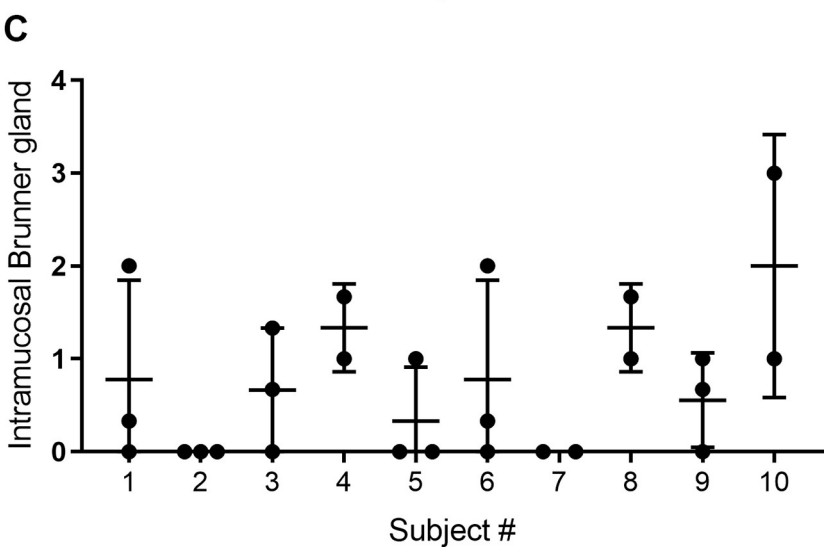

**Fig 3. Individual histology heterogeneity, Pakistani cohort.** The scores for each individual biopsy for each subject were plotted for (A) total EED histology score percent, (B) villous architecture, and (C) intramucosal Brunner glands as these showed the highest variability between biopsies. The bars indicate mean ± standard deviation.

To address this shortcoming in the field, we developed a histology index specific for duodenal biopsies that queries several distinct histologic parameters including quantity and type of various inflammatory cells and several epithelial differentiation lineage cells, mucosal architectural features, and epithelial cell injury patterns. Each parameter was given a numerical range of severity, with each numerical grade specifically defined. While the reproducibility between the three gastrointestinal pathologists in this study for grading the individual parameters varied, the overall score reproducibility was in the same range of interobserver reproducibility described for other accepted histology indices (overall kappa coefficient of 0.70), including those for grading inflammatory changes in ulcerative colitis (37) and due to *H. pylori* [52]. Thus, the performance of the histology index described here appears to conform to the general performance of histologic assessment of tissue pathology and is suitable for use in comparing cohorts with enteropathies, including EED. IELs, chronic inflammation, and villous architecture were the parameters with the lowest inter-pathologist concordance. Inflammatory markers may be better assessed by special stains, such as immunohistochemistry. Villous architecture scores also had the widest range, compared to all other parameters. Quantitative morphometry may offer more precision in characterizing this feature, but such measurements are also labor intensive and require good tissue orientation on the slide. Future work should assess whether morphometry provides substantial gains in precision compared to subjective pathologist assessments "by eye".

Our data offer provisional guidance regarding the numbers of biopsies to be obtained when attempting to identify EED. Most notably, we found no significant difference in the overall concordance between pathologists for the histologic parameters based on the reading of a single tissue fragment (Pakistani), or multiple tissue fragments to generate a composite score from the tissue on the slide (Zambian). However, the Pakistani biopsies, apportioned into separate cassettes, demonstrate considerable within-host (i.e., inter-biopsy) variability. Specifically, the degree of discordance of overall histologic scores was greater between biopsies of the same child than between those of different children in the Pakistani cohort. The greatest intra-individual variability related to the assessment of intramucosal Brunner glands, which are known to have a focal and nodular appearance when increased in the duodenum [40]. Because of its patchy nature, EED mucosal biopsy studies should include multiple biopsies, but pieces obtained can be considered as a pooled specimen set, to reduce the possibility of missing a case, to more comprehensively assess affected tissue, and to reduce analytic variability from sampling bias. This is standard endoscopic biopsy practice for diagnosing GSE [55, 56].

Our data demonstrate similarities and differences between the cohorts that warrant comment. While EED and GSE cohorts share histopathologic similarities, as suggested by prior studies [26, 40, 57–60], our index showed differences in the main histologic features that are classically associated with GSE, including intraepithelial lymphocytes. We also had the

**Table 6. Overall concordance of histologic parameters based on number of tissue fragments assessed per image.**

| Single tissue fragment | | | Multiple tissue fragments | | |
|---|---|---|---|---|---|
| Percent agreement | Fleiss' Kappa | Gwet's AC | Percent agreement | Fleiss' Kappa | Gwet's AC |
| 96.3 | 0.69 | 0.93 | 96.9 | 0.71 | 0.94 |
| 95.5 | 0.68 | 0.91 | 96.5 | 0.70 | 0.93 |

opportunity to compare a GSE case from Pakistan (excluded from this analysis) whose total histologic score was somewhat less severe than the median scores of the St. Louis GSE cohort, although overall showed a similar pattern of tissue injury (detailed comparison of scores to St. Louis GSE and Pakistani cohorts in **S4 Table**). A single case does not permit generalizations about the overlap between EED and GSE pathology, but raises the possibility that GSE disease may be distinguishable from EED by histopathology, as has been suggested in previous literature as well [26, 40, 57, 59, 60]. This may be most important in settings where both enteropathies co-exist.

There are several limitations of this current dataset. Due to ethical concerns, controls from EED endemic settings could not be included as endoscopic biopsies are invasive procedures that cannot be performed on otherwise healthy young children and clinical biopsies for non-enteropathy indications are quite rare at those sites. Along this line, while there were suggestions of cohort-specific histology features, we cannot exclude with certainty that the Zambian cohort findings are due to younger age, higher severity of the same enteropathy (namely EED), and/or contributions of a severe acute malnutrition-induced enteropathy. We attempted to perform an exploratory analysis of histopathology scores and patient characteristics but were hampered by samples sizes insufficient to detect significant associations. Such an analysis should be repeated with sufficiently powered data to test hypothesis driven approaches, and confirm or refute our findings, including those that were paradoxical (i.e., intramucosal Brunner gland density and villous architecture derangements with anthropometric Z scores) or not significant. EED might not be a globally applicable single entity. For example, there are differences between the intestinal lesions in the Pakistani and Zambian cohorts that we cannot readily explain, and might reflect genetics, environment, or the different indications for the endoscopic biopsies. Nonetheless, small bowel dysfunction appears to be a common finding in malnourished children in LMICs, and this report adds histologic data to recent attempts to better define EED for diagnostic and therapeutic purposes [9, 61]. Finally, it is also possible that as future data emerge regarding the causes and consequences of the histopathologic lesions we report and the concomitant clinical phenotypes, the entity we describe as EED might be superseded by new terminology. Nosologic reconsiderations might assign EED to be a specific enteropathy subtype, or EED itself might be subjected to its own variants.

It will be important to determine if the initial findings reported here extend to a larger and more diverse cohort of patients with EED. This scoring index is currently being applied to the approximately 550 biopsies by an expanded study team across five diverse sites: Zambia, Pakistan, Bangladesh and two North American comparison sites. This robust application will allow for refinement of the scoring system. This could include elimination of parameters that continue to demonstrate low performance in detection of EED and construction of a weighted algorithm of the index to more accurately define EED (e.g., certain parameters carry more weight toward the final score than others).

In summary, we described the development and validation of a new tool in the investigation of EED: a histology index designed for the study of small intestinal biopsies. The histology index can be used reproducibly by histopathologists to create parameter specific as well as overall pathology scores to compare to clinical and laboratory attributes in EED cohorts. The adoption of this index will aid not only in comparison of data within studies but in the comparison of data from separate studies. Given the durability of histologic specimens, there is an opportunity for this index to be used on archival samples from previous studies to provide historical comparison of cohorts in addition to those of contemporaneous and prospective studies. Perhaps most importantly, because the EED histopathology scoring index includes specific parameters indicative of injury processes, the system can be applied to identify histopathologic

processes that distinguish EED from other enteropathies and to inform novel intervention targets and assess response to interventions.

## Supporting information

**S1 Checklist. STROBE checklist.**
(DOC)

**S1 Table. Spearman correlation of total histologic score percent by pathologist pair.**
(DOCX)

**S2 Table. Median, IQR, and range of histology scores by cohort.**
(DOCX)

**S3 Table. Marginal associations with minimal effect sizes for regression analyses of histology score parameters with patient characteristics.**
(DOCX)

**S4 Table. Histology scores of Pakistan GSE case compared to Pakistan & St. Louis GSE cohorts histology score medians.**
(DOCX)

**S1 Atlas. Duodenal histopathology grading scheme used in this study.**
(PDF)

**S1 Dataset. Clinical metadata in this study.**
(CSV)

**S2 Dataset. Histopathology concordance dataset in this study.**
(CSV)

**S1 Fig. Clinical parameters, Zambian and Pakistani cohorts.** Due to differences in study design, the subjects in the Zambian cohort were (A) younger ($p = 0.0012$), (B) had diarrhea histories of longer durations ($p < 0.0001$), and (C) had borderline lower WHZ score ($p = 0.047$), Hemoglobin concentrations were marginally lower in the Zambia cohort ($p = 0.062$) and there was no difference in tTG antibody circulating concentrations (as a percent of upper limit of normal, $p = 0.16$). Statistical analysis was performed by Mann-Whitney tests.
(TIF)

**S2 Fig. Scores of individual histology features between cohorts.** None of these cohorts had significant differences in (A) neutrophilic inflammation, (B) eosinophilic infiltration, and (C) foveolar metaplasia. Sample size: Pakistani $n = 10$, Zambian $n = 16$, GSE $n = 13$, control $n = 6$.
(TIF)

**S3 Fig. Individual histology heterogeneity in the Pakistani cohort.** The scores for each individual biopsy for each subject were plotted for (A) intraepithelial lymphocytes density, (B) goblet cell density, (C) Paneth cell density, and (D) chronic inflammation. These parameters showed some within-child variation, but less than the scores for villous architecture and intramucosal Brunner glands.
(TIF)

## Author Contributions

**Conceptualization:** Syed A. Ali, M. Paul Kelly, Tahmeed Ahmed, Phillip I. Tarr, Donna M. Denno.

**Data curation:** Kelley VanBuskirk, M. Paul Kelly, Lori R. Holtz, Kamran Sadiq, Najeeha Iqbal, Beatrice Amadi, Michael H. Isaacs, Hannah Atlas, Phillip I. Tarr, Donna M. Denno.

**Formal analysis:** Ta-Chiang Liu, Kelley VanBuskirk, Christopher A. Moskaluk.

**Funding acquisition:** Syed A. Ali, M. Paul Kelly, Tahmeed Ahmed, Phillip I. Tarr, Donna M. Denno.

**Investigation:** Ta-Chiang Liu, Syed A. Ali, M. Paul Kelly, Lori R. Holtz, Omer H. Yilmaz, Kamran Sadiq, Najeeha Iqbal, Beatrice Amadi, Christopher A. Moskaluk.

**Methodology:** Syed A. Ali, M. Paul Kelly, Omer H. Yilmaz, Sana Syed, Sean Moore, I. Malick Ndao, Phillip I. Tarr, Donna M. Denno, Christopher A. Moskaluk.

**Project administration:** Syed A. Ali, M. Paul Kelly, Lori R. Holtz, Kamran Sadiq, Najeeha Iqbal, Beatrice Amadi, Sana Syed, Sean Moore, I. Malick Ndao, Michael H. Isaacs, Hannah Atlas, Phillip I. Tarr, Donna M. Denno.

**Resources:** Syed A. Ali, M. Paul Kelly, Lori R. Holtz, Kamran Sadiq, I. Malick Ndao, John D. Pfeifer.

**Software:** Michael H. Isaacs, John D. Pfeifer.

**Supervision:** Phillip I. Tarr, Donna M. Denno.

**Validation:** Ta-Chiang Liu, Kelley VanBuskirk, Christopher A. Moskaluk.

**Visualization:** Ta-Chiang Liu, Kelley VanBuskirk.

**Writing – original draft:** Ta-Chiang Liu, Kelley VanBuskirk, Phillip I. Tarr, Donna M. Denno, Christopher A. Moskaluk.

**Writing – review & editing:** Ta-Chiang Liu, Kelley VanBuskirk, Syed A. Ali, M. Paul Kelly, Lori R. Holtz, Omer H. Yilmaz, Sana Syed, Tahmeed Ahmed, Sean Moore, John D. Pfeifer, Hannah Atlas, Phillip I. Tarr, Donna M. Denno, Christopher A. Moskaluk.

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
