## [Decision Letter · Decision Letter 0]

18 Oct 2019

Dear Dr. Liu:

Thank you very much for submitting your manuscript "A novel histological index for evaluation of environmental enteric dysfunction identifies geographic-specific features of enteropathy among children with suboptimal growth" (PNTD-D-19-01300) for review by PLOS Neglected Tropical Diseases. Your manuscript was fully evaluated at the editorial level and by independent peer reviewers. The reviewers appreciated the attention to an important topic but identified some aspects of the manuscript that should be improved.

We therefore ask you to modify the manuscript according to the review recommendations before we can consider your manuscript for acceptance. Your revisions should address the specific points made by each reviewer.

(1) A letter containing a detailed list of your responses to the review comments and a description of the changes you have made in the manuscript.

(2) Two versions of the manuscript: one with either highlights or tracked changes denoting where the text has been changed (uploaded as a "Revised Article with Changes Highlighted" file ); the other a clean version (uploaded as the article file).

(3) If available, a striking still image (a new image if one is available or an existing one from within your manuscript). If your manuscript is accepted for publication, this image may be featured on our website. Images should ideally be high resolution, eye-catching, single panel images; where one is available, please use 'add file' at the time of resubmission and select 'striking image' as the file type. 

Please provide a short caption, including credits, uploaded as a separate "Other" file. If your image is from someone other than yourself, please ensure that the artist has read and agreed to the terms and conditions of the Creative Commons Attribution License at http://journals.plos.org/plosntds/s/content-license (NOTE: we cannot publish copyrighted images). 

(4) Appropriate Figure Files 

Please remove all name and figure # text from your figure files upon submitting your revision. Please also take this time to check that your figures are of high resolution, which will improve both the editorial review process and help expedite your manuscript's publication should it be accepted. Please note that figures must have been originally created at 300dpi or higher. Do not manually increase the resolution of your files. For instructions on how to properly obtain high quality images, please review our Figure Guidelines, with examples at: http://journals.plos.org/plosntds/s/figures

While revising your submission, please upload your figure files to the Preflight Analysis and Conversion Engine (PACE) digital diagnostic tool, https://pacev2.apexcovantage.com/ PACE helps ensure that figures meet PLOS requirements. To use PACE, you must first register as a user. Then, login and navigate to the UPLOAD tab, where you will find detailed instructions on how to use the tool. If you encounter any issues or have any questions when using PACE, please email us at figures@plos.org.

We hope to receive your revised manuscript by Dec 17 2019 11:59PM. If you anticipate any delay in its return, we ask that you let us know the expected resubmission date by replying to this email.

To submit your revised files, please log in to https://www.editorialmanager.com/pntd/

Sincerely,

Margaret Kosek

Associate Editor

Andrew Azman

Deputy Editor

Reviewer's Responses to Questions

**Key Review Criteria Required for Acceptance?**

**Methods**

-Are the objectives of the study clearly articulated with a clear testable hypothesis stated?

-Is the study design appropriate to address the stated objectives?

-Is the population clearly described and appropriate for the hypothesis being tested?

-Is the sample size sufficient to ensure adequate power to address the hypothesis being tested?

-Were correct statistical analysis used to support conclusions?

-Are there concerns about ethical or regulatory requirements being met?

Reviewer #1: I am satisfied with the methods presented in this paper. In some instances the sample size in insufficient to ensure power, but this is clearly stated by the authors, and the analyses are correspondingly described as 'exploratory' . Some additional analyses could be done (these are described under 'Major Comments') but if I don't feel that these should be required.

Reviewer #2: The terminology for the enteropathy in poorer countries has changed over the years and will no doubt continue to do so as the condition becomes better understood in specific populations as in this report. Referring to the condition being studied in this report as “EED” may be misleading. The term EED is generally used to describe the asymptomatic intestinal lesion that is universal in people exposed to poor sanitation and likely contributes to growth faltering in children. This is not the population studied here. Rather, these were selected groups of malnourished children who were either hospitalised with persistent diarrhoea or had not responded to standard nutritional therapy in the community. It is not known whether the intestinal lesion in these children represents the severe end of the spectrum of EED or perhaps one (or more) different gut pathologies. Indeed, figure 2 indicates clear differences in the intestinal lesion between the two malnutrition groups. It would be preferable if the terminology used for the gut lesion(s) studied in this report more closely related to the group studied rather than using “EED” as an umbrella term.

The objectives are clearly stated and the research design is appropriate given the difficulty of obtaining small intestinal biopsies in children. There is no formal sample size calculation but suffient children have been studied to assess the performance of the histology scoring system and identify major differences between the groups.

The status of the St. Louis controls is not clear. Are they children who had no GI pathology identified (i.e. “healthy” controls), or may they have had some GI pathology (e.g. reflux oesophagitis) although not GSE or a history of an inflammatory intestinal disorder? Please clarify the diagnoses, if any, in these controls – and then use terminology for this group consistently throughout the paper. The term “non-GSE” implies that only GSE was excluded.

Methods of biopsy sample handling/processing and how biopsies were orientated across the sites to ensure reliable measurement of morphometric indices should be included (especially as villous architecture was the index that varied most between histopathologists).

Conclusions based on data presented. No ethical concerns.

Reviewer #3: (No Response)

**Results**

-Does the analysis presented match the analysis plan?

-Are the results clearly and completely presented?

-Are the figures (Tables, Images) of sufficient quality for clarity?

Reviewer #1: The results are clear. Tables and Images are of sufficient quality, except for one supplemental figure (SF2) where the y-axis seems oddly scaled (possibly scaled to be equivalent to other figures, but the data are not really visible).

Reviewer #2: The abstract should report on the performance of the histology scoring system and then on the findings according to patient group.

Given that WAZ does not distinguish between wasting and stunting and that the intestinal lesion may well differ between these two manifestations of malnutrition, analysis according to WAZ is difficult to interpret and could be removed.

Reviewer #3: (No Response)

**Conclusions**

-Are the conclusions supported by the data presented?

-Are the limitations of analysis clearly described?

-Do the authors discuss how these data can be helpful to advance our understanding of the topic under study?

-Is public health relevance addressed?

Reviewer #1: Conclusions are supported by the data described, limitations are honest, and the results of the work are useful.

Reviewer #2: In the author summary, EED is referred to as a “disease”. EED is found in children with intense and repeated intestinal exposure to enteropathogens – yet, surprisingly, they remain mostly asymptomatic and without diarrhoea. It is conceivable, therefore, that EED is an appropriate adaptation to unsanitary environments that prevents frequent diarrhoea (and possibly diarrhoea mortality) albeit at the likely cost of growth faltering. Therefore, could the authors consider a term such as “condition” rather than “disease”?

Reviewer #3: (No Response)

**Editorial and Data Presentation Modifications?**

Reviewer #1: Major comments

The authors describe, “an index that will allow identification of histopathologic features unique to EED as well as those shared with other enteropathies´. It may be a question of terminology, but I question whether the tool that has been developed is ‘an EED index’, ‘a histology severity index’ or perhaps should instead be marketed as ‘a checklist’ of features to assess when examining a biopsy from a child with possible EED. This reflects the authors’ own point, in the second-to-last paragraph, that insufficient information is yet available to determine which parameters should be included in an ‘EED index’, or how much weight those parameters should be given. And, while demonstrating that certain features can be reasonably reliably assessed is worthwhile, that doesn’t really make it clear why it’s better to add those features up. Relying too heavily on summary scores can also cause things to be overlooked. So, since the more pertinent histologic features of EED still are not known, continuing to analyze separate features separately seems more prudent, for the time being.

Relatedly, it would be helpful to better understand, overall & across features, whether the Zambian and Pakistani groups demonstrated a histologic phenotype differentiating them from the GSE group, and controls. PCA and cluster analysis might help to assess the relative similarity of the groups relative to each other. Those features that appeared to best differentiate both groups from the GSE group, and from controls, could be given the greater weight within the final index. The fact that the data presented suggests that the Zambian and Pakistani groups are rather dissimilar is both interesting and mildly concerning, since it indirectly raises the question of an EED phenotype actually exists. 

As the authors themselves note, both ‘EED’ cohorts (Zambia & Pakistan) had wasting or severe wasting. The Zambian group had a further history of recent diarrhea. Therefore, it is really difficult to know to what extent observed changes to the small intestine was a result of ‘severe acute malnutrition-induced enteropathy’, versus the form of EED that might be present among children who are not wasted but stunted, or at risk of stunting. Why not re-title the paper, “… geographic-specific features of acute malnutrition-induced enteropathy…”? 

It is also mildly concerning that lactulose:rhamnose and lactulose:mannitol ratios were not associated with histologic scores. On the other hand, given that each test was performed in a subset of the data, perhaps not too much weight should be given to this finding. 

Minor comments

The statement. “EED is an underdiagnosed, highly prevalent condition among children in LMICs” in the introduction needs a citation.

In the sentence, “Based on existing data, we hypothesized that a histology index using an unbiased approach will allow identification of histopathologic features unique to EED as well as those shared with other enteropathies, and/or identify histologically-identifiable subsets of EED.” It is not clear what existing data the authors are referring to.

A few sentences of the abstract and introduction are awkwardly written. For example, the following sentence could be broken in two: “The working hypothesis is that an environmental driver, including recurrent or persistent enteric infection, infections from specific pathogen(s), an abundance of nonpathogenic fecal microbes, or distortion of intestinal microbiota composition, precipitate and/or perpetuate the small bowel response, analogous to the small bowel response to gluten in gluten-sensitive enteropathy (GSE), also known as celiac disease.”

It is unclear why the groups are described as ‘cohorts’, since, as described, only the Pakistani group appears to be part of a larger nested cohort, and the other 3 studies appear to have possibly been cross sectional. 

Supplemental Figure 2 has a strange axis, nothing can really be seen. Since no outliers extend to the upper part of the y-axis, it is unclear why the range is so large.

Further speculation about what explains differences between Zambia and Pakistan would be welcomed.

Reviewer #2: Page 8; include details of methodology and results for the lactulose:mannitol test.

Page 14: correct “Median tTG IgA concentrations as a percent of the upper limit of normal limit were …”

Fig 1: clarify the meaning of the horizontal bars with * in the legend.

Fig 2: clarify why only 8 and not 11 histology parameters are shown

Figs 1-3: it would be useful to clarify what the horizontal bars show (e.g. are these median and IQR?).

Tables 1 and 2: please keep the order of columns consistent between the tables (inc. Pakistan and Zambia).

Table 2 and 5: it is surprising that none of the GSE children had diarrhoea (and page 14); please comment. Table 2: It would be useful to add normal ranges for the lab tests in the left-hand column. Change “non-GSE control” to “Control” in footnote * (as in comment above).

Table 3: The scoring range for Villous architecture 3 and 4 and for Enterocyte injury 1 and 2 do not include the value 50% for abnormality of mucosa

Supplementary Fig. 1 E; please clarify units on y axis

Reviewer #3: (No Response)

**Summary and General Comments**

Reviewer #1: This is a useful paper that significantly advances the field of EED research. The work has been carefully done. I appreciate the effort to demonstrate inter-rater reliability as well as inter-biopsy reliability. Generally, the work leaves me scratching my head about the relationship of histological features to EED.

Reviewer #2: This is a ground-breaking study of the development and initial findings of a scoring system for the intestinal pathology in children with malnutrition. The team of international collaborators that are leading research in this area are to be congratulated for overcoming the considerable challenges of undertaking endoscopy and biopsy safely in these highly vulnerable children. The histological scoring system will be of great value in further research in this area.

It would be useful to state that there were no adverse effects of endoscopy and biopsy (unless this has already been published elsewhere).

Reviewer #3: The paper by Liu et al reports a new histological index of research and clinical value in the precise description of enteropathy. The study has the virtue of being jointly undertaken by clinicians and pathologists collaborating together. Its focus is not the general observation that  I have just made but focuses upon a very neglected area of the small intestinal enteropathy recognized to occur in children with suboptimal growth, now called environmental enteropathy within developing communities.

A comparison has been made of child patients from two communities in developing countries viz. Pakistan and Zambia with disease controls provided by child patients from St. Louis in USA with GSE , a developed community. It is not ethical to obtain small intestinal biopsies from “normal” in any of the three countries.

The more general value of this paper is that it produces evidence for the value of a comprehensive reproducible histological framework to categorize small intestinal enteropathy.

The histological data and the way it was produced from the three communities is clearly documented with aN indication of reproducibility (inter observer reproducibility), with rigorous testing. Care has been taken to harmonise the assessment from the three communities. The EED Biopsy Initiative Consortium is a fine example of international  collaborative study.

The area studied is of considerable clinical importance from the perspective of paediatric gastroenterology and beyond in the realm of nutrition.

Comparison between environmental enteropathy and GSE is important in communities such as Pakistan, where both may occur. It is important to observe that IEL histology scores for IEL were elevated for GSE in St. Louis (2.7) and in Pakistan (2.6) compared to Zambia (0.7). Elevated IEL’s are a well known histological feature of GSE but this has also been described in experimental animals (MacDonald and Ferguson 1978) and in humans (Wright et al 1977). Giardia can be recognized histologically on the surface of the mucosa of children with giardiasis, presumably this was not observed but it would be a helpful addition to the paper to have a comment that there was no histological evidence of giardia lamblia.

MACDONALD TT and FERGUSON A (1978) Small intestinal epithelial kinetics and protozoal infection in mice

Gastroenterology 74 496-500

WRIGHT SG, TOMKINS AM and RIDLEY (1977) Giardiasis: clinical and therapeutic aspects Gut 18 343 – 350k

The overall result is really just a beginning as there clearly needs to be further studies of this neglected area. These may indeed pave the way for future therapeutic endeavours.

Overall I strongly recommend publication of this paper which may indeed in the future be regarded as a seminal paper for the systematic evaluation of histology in children with small intestinal enteropathy.

PLOS authors have the option to publish the peer review history of their article (what does this mean?). If published, this will include your full peer review and any attached files.

Reviewer #1: No

Reviewer #2: Yes: Stephen Allen

Reviewer #3: No

---

## [Editor Report · Decision Letter 1]

6 Dec 2019

Dear Dr. Liu,

We are pleased to inform you that your manuscript, "A novel histological index for evaluation of environmental enteric dysfunction identifies geographic-specific features of enteropathy among children with suboptimal growth", has been editorially accepted for publication at PLOS Neglected Tropical Diseases.

Before your manuscript can be formally accepted and sent to production you will need to complete our formatting changes, which you will receive in a follow up email. Please note: your manuscript will not be scheduled for publication until you have made the required changes.

IMPORTANT NOTES

* Copyediting and Author Proofs: To ensure prompt publication, your manuscript will NOT be subject to detailed copyediting and you will NOT receive a typeset proof for review. The corresponding author will have one final opportunity to correct any errors when sent the requests mentioned above. Please review this version of your manuscript for any errors.

* If you or your institution will be preparing press materials for this manuscript, please inform our press team in advance at plosntds@plos.org. If you need to know your paper's publication date for media purposes, you must coordinate with our press team, and your manuscript will remain under a strict press embargo until the publication date and time. PLOS NTDs may choose to issue a press release for your article. If there is anything that the journal should know, please get in touch.

*Now that your manuscript has been provisionally accepted, please log into EM and update your profile. Go to http://www.editorialmanager.com/pntd, log in, and click on the "Update My Information" link at the top of the page. Please update your user information to ensure an efficient production and billing process.

*Note to LaTeX users only - Our staff will ask you to upload a TEX file in addition to the PDF before the paper can be sent to typesetting, so please carefully review our Latex Guidelines [http://www.plosntds.org/static/latexGuidelines.action] in the meantime.

Best regards,

Margaret Kosek

Associate Editor

Andrew Azman

Deputy Editor

---

## [Editor Report · Acceptance letter]

20 Dec 2019

Dear Dr. Liu,

We are delighted to inform you that your manuscript, "A novel histological index for evaluation of environmental enteric dysfunction identifies geographic-specific features of enteropathy among children with suboptimal growth," has been formally accepted for publication in PLOS Neglected Tropical Diseases.

Best regards,

Serap Aksoy

Editor-in-Chief

Shaden Kamhawi

Editor-in-Chief
